# Identification of the *Abscisic Acid-, Stress-, and Ripening-Induced* (*ASR*) Family Involved in the Adaptation of *Tetragonia tetragonoides* (Pall.) Kuntze to Saline–Alkaline and Drought Habitats

**DOI:** 10.3390/ijms242115815

**Published:** 2023-10-31

**Authors:** Hao Liu, Qianqian Ding, Lisha Cao, Zengwang Huang, Zhengfeng Wang, Mei Zhang, Shuguang Jian

**Affiliations:** 1Guangdong Provincial Key Laboratory of Applied Botany, South China Botanical Garden, Chinese Academy of Sciences, Guangzhou 510650, China; liuhao123@scbg.ac.cn (H.L.); Dingqianqian@scbg.ac.cn (Q.D.); caolisha23@mails.ucas.ac.cn (L.C.); hzw15826801754@163.com (Z.H.); wzf@scbg.ac.cn (Z.W.); 2University of Chinese Academy of Sciences, Beijing 100039, China; 3Key Laboratory of Vegetation Restoration and Management of Degraded Ecosystems, Center for Plant Ecology, Core Botanical Gardens, Chinese Academy of Sciences, Guangzhou 510650, China; 4Southern Marine Science and Engineering Guangdong Laboratory (Guangzhou), Guangzhou 511458, China; 5Center of Economic Botany, Core Botanical Gardens, Chinese Academy of Sciences, Guangzhou 510650, China; 6CAS Engineering Laboratory for Vegetation Ecosystem Restoration on Islands and Coastal Zones, South China Botanical Garden, Chinese Academy of Sciences, Guangzhou 510650, China

**Keywords:** *Abscisic Acid-, Stress-, and Ripening-Induced* (*ASR*), drought, saline–alkaline soil, water deficit, *Tetragonia tetragonoides* (Pall.) Kuntze

## Abstract

*Tetragonia tetragonoides* (Pall.) Kuntze (Aizoaceae, 2n = 2x = 32), a vegetable used for both food and medicine, is a halophyte that is widely distributed in the coastal areas of the tropics and subtropics. Saline–alkaline soils and drought stress are two major abiotic stressors that significantly affect the distribution of tropical coastal plants. Abscisic acid-, stress-, and ripening-induced (ASR) proteins belong to a family of plant-specific, small, and hydrophilic proteins with important roles in plant development, growth, and abiotic stress responses. Here, we characterized the *ASR* gene family from *T. tetragonoides*, which contained 13 paralogous genes, and divided TtASRs into two subfamilies based on the phylogenetic tree. The *TtASR* genes were located on two chromosomes, and segmental duplication events were illustrated as the main duplication method. Additionally, the expression levels of *TtASR*s were induced by multiple abiotic stressors, indicating that this gene family could participate widely in the response to stress. Furthermore, several TtASR genes were cloned and functionally identified using a yeast expression system. Our results indicate that *TtASR*s play important roles in *T. tetragonoides*’ responses to saline–alkaline soils and drought stress. These findings not only increase our understanding of the role ASRs play in mediating halophyte adaptation to extreme environments but also improve our knowledge of plant ASR protein evolution.

## 1. Introduction

*Tetragonia tetragonoides* (Pall.) Kuntze (also called New Zealand spinach or French spinach) is a halophyte and vegetable that is widely distributed in the coastal areas of tropical and subtropical regions. *Tetragonia tetragonoides* is highly adapted to seawater and drought conditions and can therefore be planted near the beach or on islands and big reefs [1]. Moreover, *T. tetragonoides* and its extracts are used in folk medicine for gastrointestinal diseases, such as gastric cancer, gastritis, gastric ulcers, and acid excess, and they may have antioxidant, antidiabetic, and anti-inflammatory effects [1,2]. The natural habitats of *T. tetragonoides* plants are usually harsh for most other plants, especially traditional crops or vegetables. To cope with these unfavorable environmental conditions, *T. tetragonoides* has adopted various resistance strategies. Related research can provide valuable information to improve abiotic stress tolerance in crops. It is therefore important to understand in depth the mechanisms by which the genome or specific genes of *T. tetragonoides* have adapted to extreme environmental abiotic stress, such as drought, saline–alkaline, and extreme high-temperature stress, to complete their life cycle.

Plants have evolved a number of different methods to deal with inevitable abiotic stress. First, the morphology of plants has developed with specialization for environmental adaptation. The increased succulence and small and glistening liquid-filled blisters in leaves and young stems enhance the salt and drought tolerance of *T. tetragonoides* in its natural habitats, and its large root system and proper creeping architecture also ensure its anti-wind capability [3]. Most importantly, as a halophyte species, the physiological and biological processes of *T. tetragonoides* in vivo, which include ion compartmentalization and homeostasis, ion extrusion, and osmotic adjustment at cellular or molecular levels, might be the key factors that determine its specific salinity stress tolerance [3,4]. Currently, research on the molecular mechanism underlying the ecological adaptability of this species is limited, and only a few reports are available [3,4,5,6]. Highly saline–alkaline soils, drought, and strong UV light or sunshine are crucial abiotic stress factors limiting the distribution of tropical coastal plants. As a special-habitat plant species, *T. tetragonoides* has adopted some special molecular and physiological mechanisms for its growth and reproduction in unusual ecological environments.

Abscisic acid (ABA)-, stress-, and ripening-induced (ASR) proteins are plant-specific families with conserved C-terminal ABA/WDS domains that play important roles in pollen or fruit ripening, abiotic stress tolerance, and the ABA signaling pathway in plants. ASRs belong to small hydrophilic proteins and are often encoded by small gene families. Since the first *ASR* gene was identified in tomato (*Solanum lycopersicum* L.) [7], an increasing number of ASR genes have been identified and cloned from different dicotyledons and monocotyledons [8,9,10,11,12]. Interestingly, no *ASR* gene members exist in the Brassicaceae family, including the model plant *Arabidopsis thaliana*. Although the ASR family has been widely studied, there have been only several reports on the function of ASRs in special-habitat plants [13,14,15,16,17,18,19]. Many genes in wild relatives provide genetic resources for germplasm improvement. More importantly, *ASR* genes playing crucial roles in the response to high salinity and drought stress in plants are of significance in plant genetic engineering.

Wild plant species are inevitably challenged by many severe environmental factors. In particular, in tropical coastal plant species, continuous salinity–alkali stress and seasonal drought, heat, UV light, and flooding are significant abiotic stressors that restrict the distribution of plants and control the productivity and alteration of generations. As an extreme halophyte, *T. tetragonoides* is a leaf succulent and can accumulate saline ions; therefore, it can grow in saline coastal regions. Undoubtedly, adaptation to salinity–alkali and drought stress in *T. tetragonoides* plants is also coupled with physiological and molecular mechanisms. *ASR* genes encode plant-specific hydrophilic proteins, which function as molecular chaperones or protective molecules in the cytosol [20] or act as transcription factors regulating gene expression in the nucleus during stress responses [21]. The involvement of *ASR*s in plant drought and salt tolerance has been documented in several species [15,16,18,19,22,23,24,25,26,27,28,29]. A reference genome sequence of *T. tetragonoides* has recently been generated. Because of the remarkable saline–alkaline and drought tolerance of this species, the identification of stress-relevant genes is important, as it may have important applications in the genetic improvement of crop resistance. In the present study, we identified *ASR* genes in *T. tetragonoides* (*TtASR*s) and characterized the structure of *TtASR*s and their chromosomal locations. We also investigated the expression profiles of *TtASR* genes in various tissues in response to different abiotic stressors and performed promoter analyses. Additionally, some *TtASR* genes were functionally identified using heterogeneous transgenic assays. Our results suggest that TtASRs may act as a protective molecule to help *T. tetragonoides* plants adapt to extreme abiotic stress in tropical and subtropical coastal regions.

## 2. Results

### 2.1. Identification of the T. tetragonoides ASR Family

Thirteen *ASR* genes were identified in the *T. tetragonoides* genome using the InterProscan search and BLAST confirmation (Table 1; Appendix A). These genes encoded proteins that had a conserved ABA_WDS domain on their C-terminus, and according to their genome locus, the genes were designated as *TtASR1*–*13*. Among these, *TtASR5* was the only reported gene that was isolated from the cDNA library of *T. tetragonoides* in our previous study [6]. Based on the similarity of their sequences, two possible segmentally duplicated gene pairs, namely *TtASR4*/*TtASR13* and *TtASR5*/*TtASR12*, were identified in *T. tetragonoides*.

### 2.2. Features of ASR Proteins

After verification of the complete open reading frame (ORF) of the full-length cDNA of each *TtASR*, the predicted amino acid sequences of the TtASRs were calculated with different biological programs. Analysis of the physicochemical properties revealed TtASRs ranging from 105 to 277 amino acids. The molecular weight varied from 12.09 to 29.1 kDa, and the pI ranged from 5.12 to 9.47. Except for TtASR6, which was neutral, seven TtASR proteins (TtASR4, TtASR5, TtASR7, TtASR9, TtASR11, TtASR12, and TtASR13) were weakly acidic, and five TtASR proteins (TtASR1, TtASR2, TtASR3, TtASR8, and TtASR10) were weakly alkaline. The instability index (II) ranged from 16.86 to 47.17, and the aliphatic index (AI) ranged from 16.68 to 62.9 (only two TtASRs, TtASR5 and TtASR12, had AI values < 40). The IIs of most of the TtASRs had values lower than 40, indicating that these proteins were relatively stable in vivo. The TtASR proteins also consisted of very high percentages of lysine (K), histidine (H), glycine (G), glutamic acid (E), and alanine (A) compared to the other amino acids (Table 1). We also predicted the disordered amino acid content of all TtASRs since some reported plant ASRs are intrinsically disordered proteins (IDPs), including SbASR-1 (*Salicornia brachiata*) [13], SlASR (*Suaeda liaotungensis*) [14], TtASR1 (wheat) [20], HvASR1 (barley) [20], and PgASR3 (pearl millet) [17]. Both TtASR5 and TtASR12 were predominantly disordered since they contained about 80% disordered amino acids. The other TtASRs showed a relatively lower disordered amino acid content, ranging from 35.85% to 56.07%. In general, the subcellular localization of most TtASRs was in the nucleus, which was predicted by different programs (Table 1). This is consistent with their biological functions as transcription factors or molecular chaperones.

### 2.3. Evolutionary Characterization and Chromosomal Locations of TtASRs

To investigate the phylogenetic relationships of TtASRs, we performed phylogenetic analysis of the TtASRs and other plant ASR proteins and established unrooted phylogenetic trees (Figure 1). All 13 TtASRs were distributed in the two larger subgroups (Figure 1A), and the chosen plant ASR proteins were classified into two subfamilies (Figure 1B). In general, TtASRs with adjacent gene localization were not evolutionarily close, such as TtASR4/TtASR5/TtASR6 and TtASR11/TtASR12/TtASR13 (Table 1), suggesting that they were not similar genes, and their functions may be highly diverse.

The evolutionary relationships among the *TtASR*s were also reflected by their chromosome maps (Figure 2). Interestingly, all 13 *TtASR*s were concentrated in the two syntenic blocks (Table 1). The encoding proteins of two gene pairs, *TtASR4*/*TtASR13* and *TtASR5*/*TtASR12*, were highly homologous but distributed on different chromosomes.

### 2.4. Gene Structures and Protein Motif Compositions

To better understand the structural features of *TtASR*s, gene structure analyses were performed using the GSDS tool. The results reveal that each *TtASR* member had two exons, ranging from 147 to 642 bp in length, and one intron ranging from 210 to 1641 bp in length (Figure 3A). The gene structures of all *TtASR*s were generally considered conservative.

Protein motifs are critical for biochemical functions. The conserved motif analyses were executed with the MEME tool with motifs 1–10 (Figure 3B). All TtASR proteins contained conserved ABA/WDS domains (motifs 1, 2, and 4) and a small N-terminal consensus containing a His-rich sequence (motif 3). Motif 8 was present in the N-terminal of TtASR11 and TtASR12, and motif 10 was present only in the C-terminus of TtASR9 and in the near N-terminus middle part of TtASR12. The other four motifs (5, 6, 7, and 9) were specific to TtASR5 and TtASR12, which were composed mainly of Gly-rich sequences.

### 2.5. Cis-Acting Regulatory Elements

The promoter sequences in the 2 kb region upstream of all *TtASR*s’ 5′-untranslated regions (5′-UTRs) were analyzed. Based on their predicted functions, the 17 identified *cis*-acting elements were classified into three groups, namely hormone response, stress response, and transcript factor (TF) binding sites (Figure 4, Appendix A). The hormone response elements included salicylic acid (SA) response (TCA) and abscisic acid (ABA) response (including ABRE, ABRE3a, and ABRE4). The stress response elements included low-temperature-responsive (LTR) and defense- and stress-responsive (TC-rich repeat) stress response elements (STREs). The TF binding sites included MYB (MYB core element, involved in flavonoid biosynthesis, etc.), the MYB binding site involved in light responsiveness (MYB binding site) or other stress responses (MYB recognition site, MYB-like sequence), MYB binding site involved in drought-inducibility (MBS), MYBHv1 binding site (CCAAT-box), and MYC (dehydration responsive) or Myc (other abiotic stress responsive). The analysis revealed that the promoter regions of the TtASR genes’ promoter regions contained a large number of possible *cis*-acting elements, with a total of 231 elements identified. Moreover, most of the *TtASR* genes contained TF binding sites, such as MYB binding site elements and Myc-binding elements. Additionally, the ABA response elements (ABRE, ABRE3a, and ABRE4) were also detected in most *TtASR* promoters. These findings suggest that the *TtASR* genes may participate in many complex regulatory pathways involved in plant environmental stress responses.

### 2.6. Expression Profiles of TtASRs in Different Tissues and Plants in Response to Different Stressors

Tissue-specific expression profiles of *TtASR*s were analyzed via RNA-Seq in the root, stem, leaf, flower bud, and young fruit of *T. tetragonoides* plants. Overall, four *TtASR*s, namely *TtASR4*, *TtASR7*, *TtASR11*, and *TtASR12*, were highly expressed in all five tissues, and *TtASR7* almost presented a constitutively high expression pattern in all five tissues (Figure 5). *TtASR2*, *TtASR3*, *TtASR6*, and *TtASR10* showed obviously lower expression levels in all five tissues compared to the other *TtASR*s. This expression profile indicates that different *TtASR* members play specific roles in the regulation of *T. tetragonoides* development by regulating its expression levels.

We also performed gene expression analysis in different *T. tetragonoides* tissues under various stress challenges, including heat (45 °C), salt (600 mM NaCl), alkalis (150 mM NaHCO_3_, pH 8.2), and high osmotic stress (drought simulated with 300 mM mannitol), mainly based on the natural habitat of *T. tetragonoides*. Heat stress for *T. tetragonoides* plants was only continued for 2 h because longer treatment (45 °C for 2 h) led to obvious wilting. The other three stress challenges lasted 2 days (48 h). The RNA-Seq results show that under the heat (physical) challenge, the expression changes in *TtASR*s were more distinct in the roots and leaves than in the stems. *TtASR2*, *TtASR3*, *TtASR4*, *TtASR5*, *TtASR10*, and *TtASR13* presented induced expression changes in at least two tissues. When challenged by salt, alkali, or drought (chemical) stress, the expression changes in *TtASR*s seemed to be more distinct in the aerial parts of the *T. tetragonoides* plants (leaves and stems) than in the underground parts (roots), indicating that these challenges caused injury to root tissues and inhibited the activities of cells, especially in the root samples under 48 h stress treatments (Figure 6). Overall, the expression of *TtASR3*, *TtASR4*, *TtASR5*, *TtASR11*, *TtASR12*, and *TtASR13* clearly induced transcriptional patterns in at least two tissues under these stress challenges (Figure 6).

The expression profiles of stress-responsive *TtASR*s were also validated via qRT-PCR. Based on the previous RNA-Seq results, several candidate genes, including *TtASR1*, *TtASR4*, *TtASR7*, *TtASR9*, *TtASR11*, and *TtASR12*, were chosen, and the treatment time points were refined to 0, 2, 8, and 24 h. In the root sample, salt/alkalis caused obvious induced expression of *TtASR1*, *TtASR7*, *TtASR9*, and *TtASR12*, while heat upregulated the expression of *TtASR7* and *TtASR9*. In the stem sample, *TtASR1*, *TtASR9*, and *TtASR12* presented elevated expressions under four stress challenges, and heat seemed to cause the most change. In the leaf sample, the expression changes in *TtASR1*, *TtASR4*, *TtASR9*, and *TtASR12* were the most significant (Figure 7).

### 2.7. Abiotic Stress Tolerance of Yeast Heterologously Expressing TtASRs

We performed the functional identification of several candidate *TtASR*s with a yeast heterologous expression system. In brief, six *TtASR*s, namely *TtASR1*, *TtASR4*, *TtASR7*, *TtASR9*, *TtASR11*, and *TtASR12*, were cloned and functionally verified under different challenges. Following the standard gene cloning and construction procedures for generating the expression vectors TtASRs-pYES2, the target genes were transformed into different yeast strains, including the WT and several mutants (*ycf1∆* sensitive to Cd, *smf1∆* sensitive to Cd, *cot1∆* sensitive to Co, *cup2∆* sensitive to Cu, *pmr1∆* sensitive to Mn, and *zrc1∆cot1∆* sensitive to Zn).

To gain insight into the functional significance of these six *TtASR*s in salt tolerance, we explored the effect of NaCl on the activities of yeast expressing different *TtASR*s and empty vector pYES2 (as a negative control). As shown in Figure 8, all six *TtASR*s improved the salt tolerance of yeast, and yeast expressing *TtASR1*, *TtASR7*, and *TtASR9* seemed to survive well, even at a high NaCl concentration (1.25 M). Yeast-transformed pYES2 presented weak viability at both 1 and 1.25 M NaCl compared with yeast expressing six *TtASR*s.

We also tested the heavy metal tolerance of yeast expressing six *TtASR*s. The *ycf1∆* strain expressing *TtASR4* grew well under low Cd concentration stress (30 μM), while even under a higher Cd stress (40 μM), the Cd tolerance was completely inhibited (Figure 9A). Similar results were also observed in *smf1∆* and *zrc1∆cot1∆* mutant strains (Figure 9B,C). *TtASR1* and *TtASR11* slightly elevated the Ni tolerance of *smf1∆* (2 mM Ni), and *TtASR1*, *TtASR4*, *TtASR7*, and *TtASR11* slightly elevated the Zn tolerance of *zrc1∆cot1∆* (0.4 mM Zn). However, for the other three heavy-metal-sensitive yeast mutant strains, namely *cot1∆*, *cup2∆*, and *pmr1∆*, none of the six tested *TtASR*s exhibited signs of elevated Co, Cu, and Mn tolerance for yeast cell activities (Figure 9D–F).

## 3. Discussion

*Tetragonia tetragonoides* is an excellent wild vegetable resource that has the concomitant functions of both medicine and foodstuff and is also used as a useful landscape plant for the restoration of the ecological environment of some tropical coastal regions and coral islands. This plant species presents extreme saline–alkaline and drought resistance, and understanding its tolerance mechanism will open up opportunities to apply this knowledge to improve abiotic stress tolerance in crops for future genetic improvement. The *ASR* gene family is plant-specific, functioning in plant resistance to unfavorable stimuli, and it has been suggested that it plays multiple roles in plants during plant development and evolution [9]. Since the first isolation of *SlASR* from tomato (*Solanum lycopersicum*), ripe fruit, and water-stressed leaf cDNA libraries in 1993 [7], increasing evidence has suggested a protective function for ASR proteins against high-salinity or desiccation [30,31] and other stressors [32]. And plant *ASR*s have been considered as metal-responsive genes and are involved in plant detoxification of some heavy metals [33,34]. Plant *ASR*s are widely distributed in many species and have been reported to be involved in the response of plants to various abiotic stressors [9,35], leading to great interest in *ASR*s as promising genes for the genetic improvement of crop abiotic stress tolerance.

As stress-inducible genes, most *ASR*s have been cloned and reported in some crops [10,11,12,16,17,21,23,26,27,28,29,30,31,32,33,34] and several wild plants [13,14,15,22,24,25], mainly being involved in salt, drought, cold, and oxidative stress, disease resistance, ABA responses, and other developmental or metabolism processes. However, there is little information on *ASR* genes in halophytes, and only specific *ASR* members have been characterized in special-habitat plants, including halophytes [13,14,18,19,36]. *Tetragonia tetragonoides* is widely distributed in coastal sandy fields of the tropics and subtropics, possessing the typical characteristics of drought, salinity/alkali, and heat tolerance, disease resistance with medical nutrition, and high economic value [37]. Inevitably, there is a strong or moderate association between typical stress-related genes (such as *ASR*s) and plant stress adaptability (such as drought and salinity/alkali tolerance). To further expand this knowledge, in the present study, we comparatively investigated the sequences and expression patterns of the *TtASR* gene family in wild *T. tetragonoides*, providing further insights into the possible roles of *TtASR*s in this special-habitat species adapted to tropical coastal regions.

The *ASR* family is a small and relatively conserved family across the plant kingdom, although it is missing from the Brassicaceae family. The availability of genome sequencing is a driving force for the investigation of these functional genes. Several *ASR* families from different plant species have been systematically characterized, such as in monocotyledon banana (Musaceae, at least four members) [8], maize (nine *ZmASR*s) [34], wheat (hexaploid, thirty-three *TaASR*s) [11,38], foxtail millet (six *SiASR*s) [24], *Brachypodium distachyon* (Poaceae, five *BdASR*s) [25], dicotyledon tomato (Solanaceae, five *SlASR*s) [30], common bean (*Phaseolus vulgaris* L., at least two *PvASR*s) [39], apple (Rosaceae, five *Md-ASR*s) [32], and gymnosperm loblolly pine (*Pinus taeda* L., four *PtASR*s) [36]. Except for hexaploid wheat, with 33 *TaASR* members, all the other above-reported diploid species possess no more than 10 *ASR* members. In *T. tetragonoides*, 13 *TtASR*s were characterized, and some gene pairs showed obvious expansion, such as *TtASR4*/*TtASR13* and *TtASR5*/*TtASR12*. Accordingly, the genome size of *T. tetragonoides* is moderate, at about 400 Mb (16 pairs of chromosomes). Plants growing in tropical coastal regions are often subject to drought, high salt/alkali stress, and heat. The different rates of gene gain and loss can cause the copy number of homologous gene groups to vary among species, which provides a genetic basis for phenotypic adaptation, species diversification, and even genome size adjustment. In contrast, the obvious expansion of the number of *TtASR*s in the *T. tetragonoides* genome may be the result of the adaptive evolution in response to special habitats with multiple stress challenges, especially for high-salinity and water-deficient habitats.

In our study, phylogenetic analysis divided TtASR proteins into two major groups with distinctive evolutionary relationships (Figure 1A). We also identified, via conserved motif analysis, 10 motifs in these proteins (Figure 3B), which presented highly conservative features, especially in the ABA/WDS domain (Figure 3B). Clearly, the gene structure of TtASRs is also relatively conservative, and all TtASRs have two exons and one intron. The conservation in the sequence and structure of TtASRs indicates that the main functions were relatively conserved and presented little diversity. Therefore, to a certain extent, this has reflected the important biological functions of TtASRs as stress regulators, which are also relatively conserved or maybe consolidated.

However, the biological functions of plant *ASR*s can be regulated in multiple ways, mainly through transcriptional control, based on their strength or absence. Previous studies have shown that plant *ASR*s may be involved in the response to various stressors and related signaling molecules at the transcript level [9,35]. The *cis*-acting regulatory elements in promoter regions are the key factors that initiate transcriptional responses to various signals and stimuli. Our analysis of the *TtASR* promoter regions indicated that most *TtASR* promoters contain rich *cis*-acting elements in response to phytohormones and light. ABREs (including ABRE3a and ABRE4), as major *cis*-acting elements in ABA-responsive gene expression, were the most abundant *cis*-acting elements. The ABA signaling pathway is central to stress responses in plants. Moreover, most *TtASR*s had multiple stress-responsive elements, such as the MYB transcription factor-regulating site (including the MYB core, MYB binding site, MYB recognition site, MYB-like sequence, MRE for light responsiveness, and MBS involved in drought stress), MYC transcription factor-regulating site (including the MYC core and Myc), and TC-rich repeats (involved in defense and stress responsiveness) (Figure 4). The frequent occurrence of these stress-related *TtASR* promoters further suggests a close relationship between the *TtASR* family and the plant stress response. This also implies the essential roles of *TtASR* family genes in drought and salt/alkali adaptability.

To gain further insights into the function of *TtASR*s in developmental regulation and stress tolerance in the growth cycle of *T. tetragonoides*, the expression profiles of *TtASR*s were investigated using RNA-Seq (Figure 5 and Figure 6) combined with qRT-PCR (Figure 7). In general, tissue- and development-specific ASR accumulation in plants is universal, especially when this kind of protein is named “ripening-induced”. ASRs also belong to numerous desiccation-tolerant proteins identified in vivo. The relative expression levels of the tomato *ASR* genes presented obvious differences in vegetative and reproductive tissues, as well as being differentially responsive to ABA and abiotic stress [30]. For example, the transcriptional profile of rice *ASR* genes in different tissues seemed to be specific to certain tissues and organs [31]. The six *SiASR*s in foxtail millet were expressed in both roots and leaves, although their expression profiles were predominantly differentiated [24]. In our study, the *TtASR* transcripts were localized more in mature stems and leaves than in young roots, flowers, and fruits (Figure 5), indicating their regulatory roles in development, as well as their possible protective roles in stress and water deprivation sensing. Water deprivation can be caused by multiple factors in plants; however, for tropical coastal-growing *T. tetragonoides*, these factors are nothing more than direct sunlight (oxidative stress), dramatic temperature fluctuations (heat), and osmotic pressure (ion stress or drought).

To further clarify the function of *TtASR*s under abiotic stress conditions, *TtASR*s were ectopically expressed in the yeast system (Figure 8 and Figure 9). Our results for yeast growth under different challenges indicate that these *TtASR*s exercise their protective roles in yeast cells, both under high salinity and under heavy metal toxicity. Many studies have demonstrated that ASR proteins play important roles in halophytes or drought-enduring plant responses to multiple abiotic stressors, especially salt and drought tolerance [9,11,13,14,15,16,17,35,40]. Most of this research was carried out with a plant heterologous transgenic system, such as in Arabidopsis [14] or tobacco [13], while some research was performed with bacteria [15] or yeast [6,27,34], both of which have shorter growth cycles and more obvious phenotypes under challenges. To quickly assess *TtASR*s’ roles in stress tolerance, a yeast heterologous expression system was used. Heterologous expression of all six *TtASR*s could improve the salt tolerance of yeast cells (Figure 8), while the extent to which it improved differed among different members. This difference may also reflect the specificity and adjustability of the *TtASR*s’ functions.

Plant ASRs can also act as metalloproteins, which may be due to their possible N-terminal metal-binding domain, which is composed of a stretch of at least six His residues [33,34,41], or other typical amino acid stretches, such as “PEHAHKHK” [20,42]. To date, there have only been a few reports on plant *ASR*s related to metal stress [33,34,42,43]. In the present study, the possible metal tolerance of six *TtASR*s was evaluated with a series of metal-sensitive yeast mutant strains (Figure 9), including cadmium-sensitive yeast mutant *ycf1Δ*, cobalt-sensitive *cot1Δ*, copper-sensitive *cup2Δ*, manganese-sensitive *pmr1Δ*, nickel-sensitive *smf1Δ*, and zinc-sensitive double-mutant *zrc1Δcot1Δ*. Interestingly, the metal chelation or tolerance mediated by different *TtASR*s was also member-specific (Figure 9), which was still somewhat similar to the results of *ZmASR*s [34]. Compared with other plant-specialized metal detoxification proteins, such as metallothionein [44] or phytochelatin synthase [45], the plant ASRs’ detoxification capability was perhaps slightly inferior. Many halophytes also accumulate metals and are planted for their phytoremediation potential [46,47,48]. Being a native tropical coastal species, *T. tetragonoides* can survive under a mixture of salt and toxic metal ions, whereas most glycophytes cannot, thus giving them the potential ability to perform phytoremediation, particularly in heavy-metal-polluted soils in offshore areas. It is important to explore the possible biochemical mechanisms that enable *T. tetragonoides* to thrive under toxic levels of salinity and heavy metals. Undoubtedly, the tolerance of *TtASR*s to high salinity and multiple metals gives this species the potential for compound ion toxicity. More research is needed to investigate the effects of these and other related economic uses of *T. tetragonoides*.

## 4. Materials and Methods

### 4.1. Identification of TtASR Genes in T. tetragonoides

The *T. tetragonoides* genome was completely sequenced, and proteins were noted with InterProscan (https://www.ebi.ac.uk/interpro/search/sequence/ (accessed on 1 March 2023)) to assess the conserved domains and motifs (e < 1 × 10^−5^). The ABA_WDS domain (Pfam No. PF02496 or InterPro No. IPR003496) was searched as a model, and the protein sequences containing this domain were screened using HMM3.0 software. The domains were then confirmed using the NCBI CDD program (https://www.ncbi.nlm.nih.gov/structure/cdd (accessed on 1 March 2023)).

### 4.2. Multiple Sequence Alignment and Phylogenetic Analysis of TtASR Family Proteins

Multiple sequence alignments of all ASR proteins (*Solanum lycopersicum* [30], *Oryza sativa* [31], *Brachypodium distachyon* [25], *Glycine max* (G.max Wm82.a2.v1; https://phytozome-next.jgi.doe.gov/ (accessed on 1 March 2023), *Malus × domestica* [32], and *Pinus taeda* [36]) were performed using ClustalW software by MEGA (version 6.06), and a phylogenetic tree was constructed using the neighbor-joining (NJ) phylogenetic method in MEGA 6.0 with 1000 bootstrap replicates. *TtASR*s were mapped on *T. tetragonoides* chromosomes according to the positional information of the *TtASR* genes in the *T. tetragonoides* genome database and were displayed using MapInspect software (https://mybiosoftware.com/mapinspect-compare-display-linkage-maps.html (accessed on 1 March 2023)).

### 4.3. Analysis of Gene Structures and Conserved Protein Motifs

The exon–intron structures of the *TtASR* genes were analyzed using the web server GSDS (Gene Structure Display Server, http://gsds.cbi.pku.edu.cn/ (accessed on 1 March 2023)) based on the comparison of cDNA sequences with their genomic DNA sequences. The subcellular localization of each TtASR protein was predicted using the online tool WoLF_PSORT (https://www.genscript.com/wolf-psort.html (accessed on 1 March 2023)) and the Plant-mPLoc server (http://www.csbio.sjtu.edu.cn/bioinf/plant-multi (accessed on 1 March 2023)). The 3D structures of the TtASRs were analyzed using the Phyre^2^ program (http://www.sbg.bio.ic.ac.uk/phyre2/html/page.cgi?id=index (accessed on 1 March 2023)). The motifs of TtASR proteins were predicted using MEME (http://meme-suite.org/index.html (accessed on 1 March 2023)), with the maximum number of motifs set to 10 and the optimum motif width set to 20–50 residues. The identified TtASR protein sequences were used to compute the MW and pI of proteins using Expasy tools (https://web.expasy.org/protparam/ (accessed on 1 March 2023)).

### 4.4. Promoter Sequence Profiling of TtASRs

Putative *TtASR* promoter sequences (2000 bp upstream of ATG) were retrieved from the *T. tetragonoides* genome database (Appendix A). Sequences were then uploaded to the PlantCARE database (http://bioinformatics.psb.ugent.be/webtools/plantcare/html/ (accessed on 1 March 2023)) for *cis*-acting regulatory element analysis. The *cis*-acting elements were classified as hormone specific (gibberellin-responsive elements, MeJA-responsive elements, salicylic acid-responsive elements, abscisic acid response elements, and ethylene response elements), abiotic stress responding (low-temperature response elements, elicitor response elements, defense and stress response elements, stress response elements, and pathogen- or wounding- inducible elements), and transcript factor binding sites (MYB and MYC binding sites). The elements were summarized, and *TtASR*s’ promoters were visualized using TBtools 2.003 [49].

### 4.5. Plant Materials and Stress Treatments

*Tetragonia tetragonoides* plants growing in the South China Botanical Garden (SCBG, 23°18′76″ N, 113°37′02″ E) were used in this study. To analyze the tissue-specific transcriptional patterns of the identified *TtASR*s, roots, stems, leaves, flowers, and fruits were gathered from *T. tetragonoides* plants grown in the SCBG. To analyze the expression patterns of the *TtASR*s responding to various stressors, 60-day-old *T. tetragonoides* seedlings were exposed to different stressors. In brief, for heat stress, seedlings were moved to a 45 °C illumination incubator. For the high-osmotic-stress treatment, seedlings were removed from their pots, carefully washed with water to remove soil from the roots, and transferred to a 300 mM mannitol solution. For high salt stress, the seedlings were soaked in a 600 mM NaCl solution. For high alkali stress, the seedlings were soaked in a 150 mM NaHCO_3_ solution (pH 8.2). The seedling roots, stems, and young leaves from the *T. tetragonoides* plants were collected at 0, 2, and 48 h during the previously described stress treatments, with the 0 time point used as the control. All samples were immediately frozen in liquid nitrogen and stored at −80 °C for subsequent gene expression analysis. Three independent biological replicates were used.

### 4.6. Expression Analysis of TtASRs

A transcriptome database was constructed for *T. tetragonoides* using Illumina HiSeq X sequencing technology. The quality of the RNA sequencing (RNA-Seq) datasets created from five tissues (roots, stems, young leaves, flowers, and young seeds collected from *T. tetragonoides* growing in SCBG) was examined using FastQC (http://www.bioinformatics.babraham.ac.uk/projects/fastqc/ (accessed on 1 March 2023)), which produced 40 Gb clean reads. Clean reads were mapped to the *T. tetragonoides* reference genome using Tophat v.2.0.10 (http://tophat.cbcb.umd.edu/ (accessed on 1 March 2023)). Gene expression levels were calculated as fragments per kilobase of transcript per million mapped reads (FPKM) according to the length of the gene, and the read counts were mapped to the gene, as follows: FPKM = total exon fragments/[mapped reads (millions) × exon length (kb)]. The expression levels (log2) of *TtASR*s were visualized as clustered heatmaps using TBtools.

The transcript abundance of several *TtASR*s’ transcripts was also investigated using qRT-PCR. In brief, total RNA was extracted from *T. tetragonoides* seedling tissues under stress treatments and reverse-transcribed to cDNA. Total RNA samples were isolated using the EasyPure^®^ Plant RNA Kit (TransGen Biotech, Beijing, China), the RNA was quantified using NanoDrop1000 (NanoDrop Technologies, Inc., Wilmington, DE, USA), and the integrity was evaluated on 0.8% agarose gel. The cDNA was synthesized using cDNA Synthesis SuperMix (TransGen Biotech, Beijing, China) according to the manufacturer’s instructions. Quantitative reverse transcription PCR (qRT-PCR) was conducted using the LightCycler480 system (Roche, Basel, Switzerland) and TransStart Tip Green qPCR SuperMix (TransGen Biotech, Beijing, China). All gene expression data obtained via qRT-PCR were normalized to the expression of *TtACT* (NCBI accession No.: MH33308). The primers used for qRT-PCR (TtACTRTF/TtACTRTR for the reference gene and other TtASR-specific primer pairs) are listed in Appendix A.

### 4.7. In Vivo Stress Tolerance Assay for TtASR Overexpression in Yeast

The ORFs of several *TtASR*s were PCR-amplified from different cDNA sample templates of *T. tetragonoides* with gene-specific primer pairs (Appendix A). The PCR fragments of the ORFs were then purified and cloned into the *Bam*HI and *Eco*RI sites of pYES2 to yield recombinant plasmids of TtASRs-pYES2 and sequenced. The recombinant plasmid TtASRs-pYES2 and the empty vector pYES2 (a negative control) were transformed into wild-type (WT) Saccharomyces cerevisiae (BY47471; MATa; *his3Δ1*; *leu2Δ0*; *met15Δ0*; *ura3Δ0*) and several heavy-metal-sensitive mutant strains *cot1Δ*, *cup2Δ*, *pmr1Δ*, *smf1Δ*, and *ycf1Δ* and double-mutant strain *zrc1Δcot1Δ* (BY4741; MATa; *his3Δ1*; *leu2Δ0*; *met15Δ0*; *ura3Δ0*; *zrc1::natMX cot1::kanMX4*). The WT (Y00000) and mutant strains *cot1Δ* (Y01613), *cup2Δ* (Y04533), *pmr1Δ* (Y04534), *smf1Δ* (Y06272), and *ycf1Δ* (Y04069) were obtained from Euroscarf (http://www.euroscarf.de/index.php?name=News (accessed on 1 March 2023)). The double-mutant strain *zrc1Δcot1Δ* was provided by Yuan et al. [50]. Plasmids were introduced into yeast strains using a standard polyethylene glycol (PEG)–lithium acetate-based transformation protocol. The yeast spot assay for NaCl and heavy metal tolerance was performed as previously described [6].

### 4.8. Statistical Analysis

All experiments in this study were repeated three times independently, and the results are shown as the mean ± standard deviation (SD) (n ≥ 3). Pairwise differences between means were analyzed using Student’s *t*-tests in Microsoft Excel 2010.

## 5. Conclusions

In the present study, 13 *TtASR* genes were identified in the *T. tetragonoides* genome and divided into two clades. A phylogenetic tree was constructed to confirm the relationships between TtASRs and other plant ASR proteins. Gene duplication plays a pertinent role in generating gene families in the process of evolution. The expansion of the *TtASR* gene family mainly occurred through segmental duplication. The conservation and diversity of genes in this gene family were also illustrated with gene exon/intron structure analysis and protein motif composition. Many *cis*-acting elements involved in stress, development, and hormone response were identified in the promoter regions of *TtASR* members, suggesting that *TtASR*s may participate in the response to various abiotic stressors and in growth regulation, thereby mediating the plant’s adaptability to tropical coastal areas. RNA-Seq data and qRT-PCR analysis results demonstrate that *TtASR*s were widely expressed in different *T. tetragonoides* tissues, with the highest expression present in the stems and leaves; some *TtASR*s were obviously induced by stress challenges. This study provides important clues for future research on the *TtASR* family in *T. tetragonoides* and sheds light on the molecular regulatory mechanisms underlying stress responses and environmental adaptability.

## Figures and Tables

**Figure 1 ijms-24-15815-f001:**
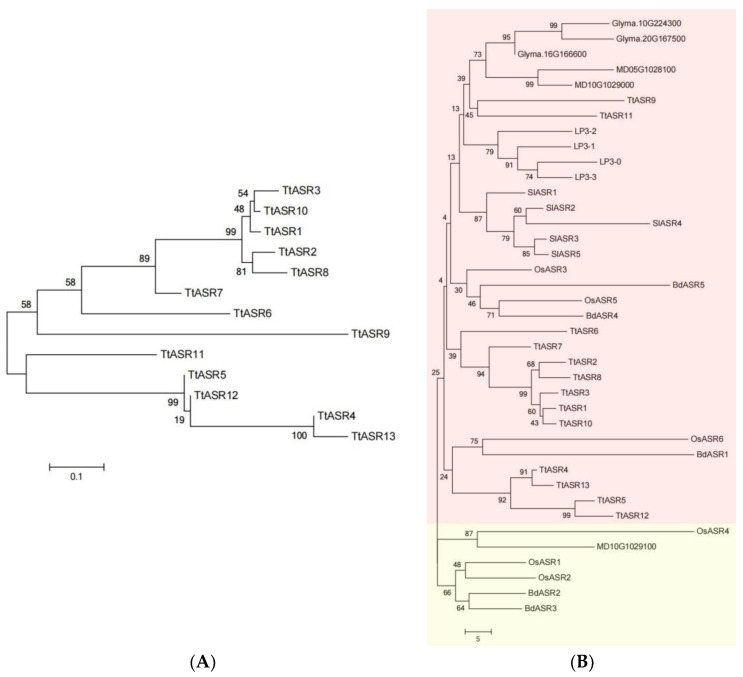
(**A**) Phylogenetic relationships of the 13 TtASRs from *T. tetragonoides*. (**B**) Phylogenetic relationships of TtASRs and other plant ASRs. The different colors show different subgroups. The phylogenetic tree is constructed using MEGA 6.0 software, with ClustalW alignment, the neighbor-joining (NJ) method, the bootstrap method, and 1000 repetitions.

**Figure 2 ijms-24-15815-f002:**
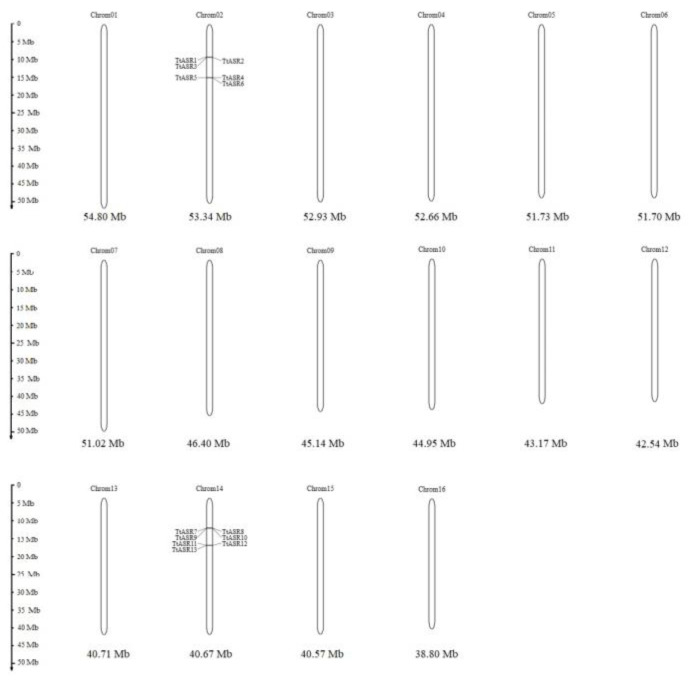
Locations of the 13 *TtASR*s on 16 chromosomes of *T. tetragonoides*.

**Figure 3 ijms-24-15815-f003:**
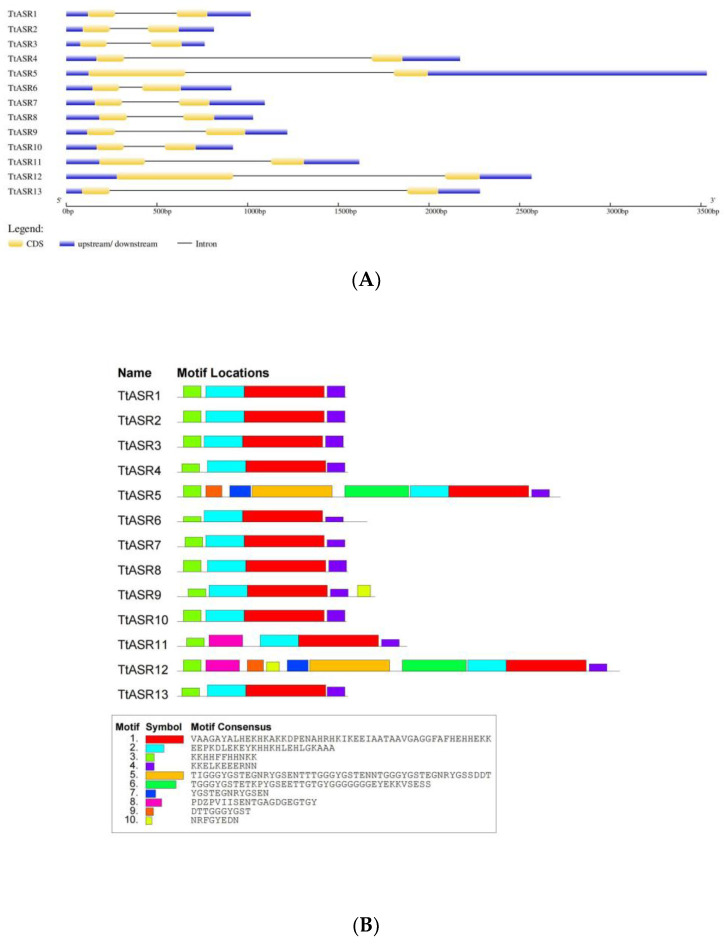
The genes’ structure and protein motif compositions of the TtASRs in *T. tetragonoides*. (**A**) The exon–intron organization of the *TtASR*s was constructed using GSDS 2.0; (**B**) the conserved motifs of each TtASR were identified using the MEME web server. Different motifs are represented by differently colored boxes.

**Figure 4 ijms-24-15815-f004:**
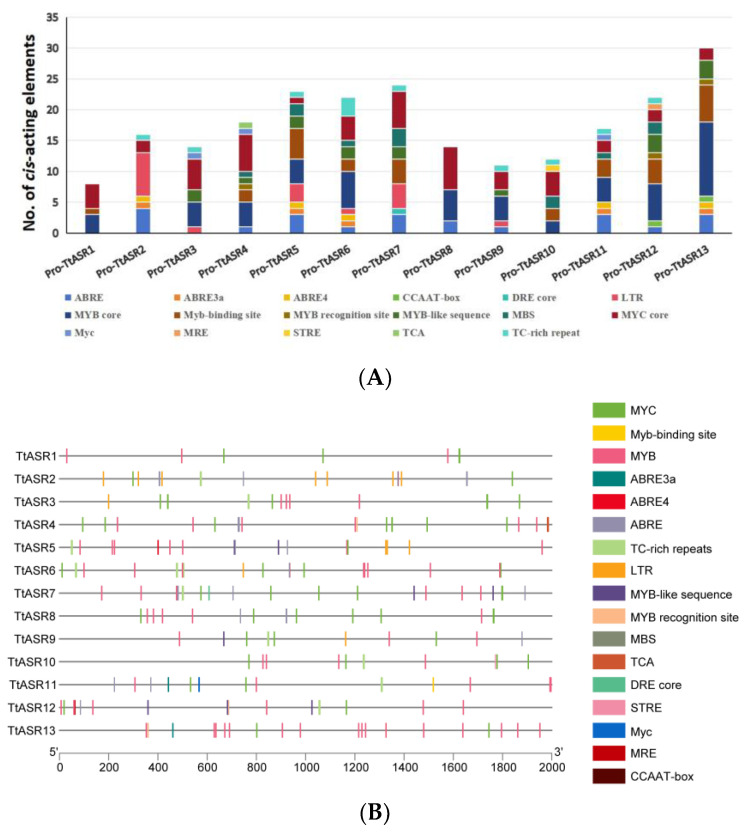
Numbers and distribution of the *cis*-acting elements in the *TtASR*s’ promoter regions. (**A**) Summaries of the seventeen *cis*-acting elements in the *TtASR*s’ promoter regions; (**B**) distribution of the seventeen *cis*-acting elements in the *TtASR*s’ promoter regions. The elements are represented by different symbols. The scale bar represents 200 bp. The detailed information of all *cis*-acting elements in the *TtASR*s’ promoter regions is listed in Appendix A.

**Figure 5 ijms-24-15815-f005:**
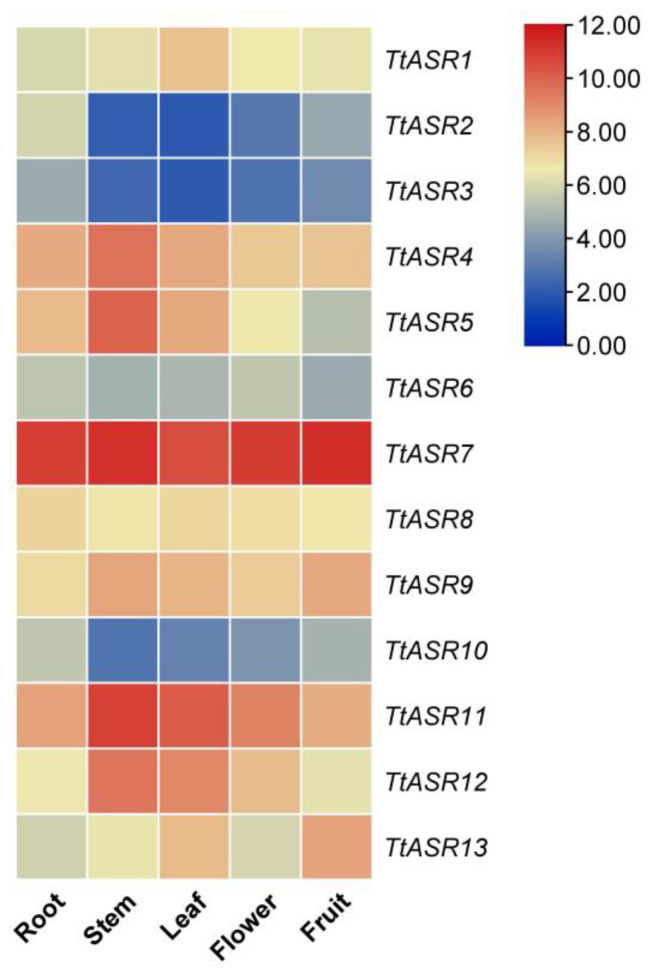
Heatmaps showing the expression levels of the *TtASR*s in the root, stem, leaf, flower bud, and young fruit of *T. tetragonoides* plants. The expression level of each gene is shown in FPKM (log2). Red denotes high expression levels, and blue denotes low expression levels.

**Figure 6 ijms-24-15815-f006:**
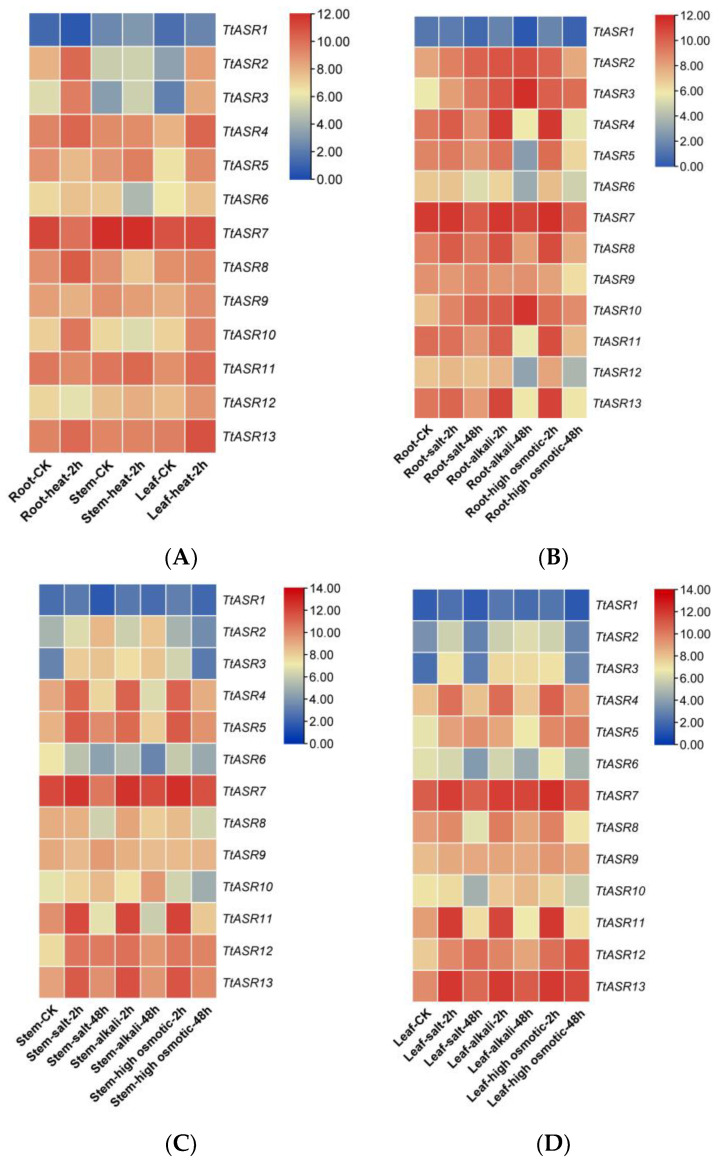
Heatmaps showing (**A**) the expression patterns of the *TtASR*s under heat; (**B**–**D**) the expression patterns of the *TtASR*s under different high-salinity (600 mM NaCl), high-alkali (150 mM NaHCO_3_), and high-osmotic-stress (300 mM mannitol) challenges, where the figures are separated according to the different *T. tetragonoides *tissues, including roots (**B**), stems (**C**), and leaves (**D**). The expression level of each gene is shown in FPKM (log2). Red denotes high expression levels, and blue denotes low expression levels.

**Figure 7 ijms-24-15815-f007:**
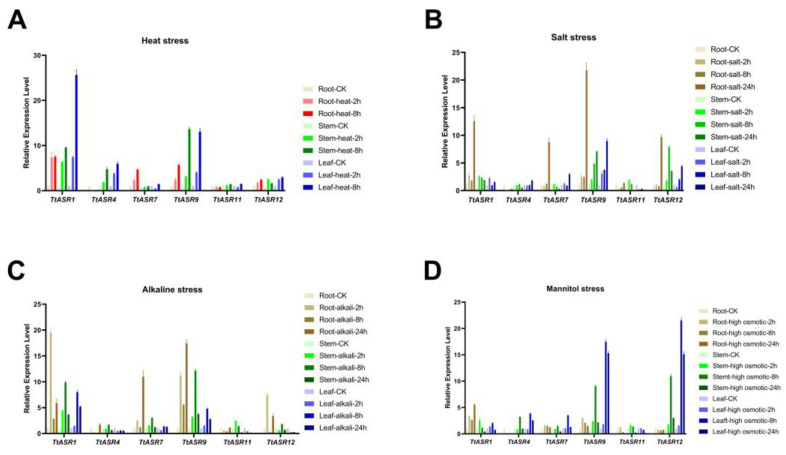
Quantitative RT-PCR detection of the expression levels of the selected *TtASR*s (*TtASR1*, *TtASR4*, *TtASR7*, *TtASR9*, *TtASR11*, and *TtASR12*) when responding to different stresses, including heat, high salinity, high alkaline, and high osmotic stress in *T. tetragonoides* seedling plants. (**A**) Parameters: 45 °C heat; (**B**) 600 mM NaCl; (**C**) 150 mM NaHCO_3_; (**D**) 300 mM mannitol. Relative expression values were calculated using the 2^−ΔCt^ method with housekeeping gene *TtACT* as a reference gene.

**Figure 8 ijms-24-15815-f008:**
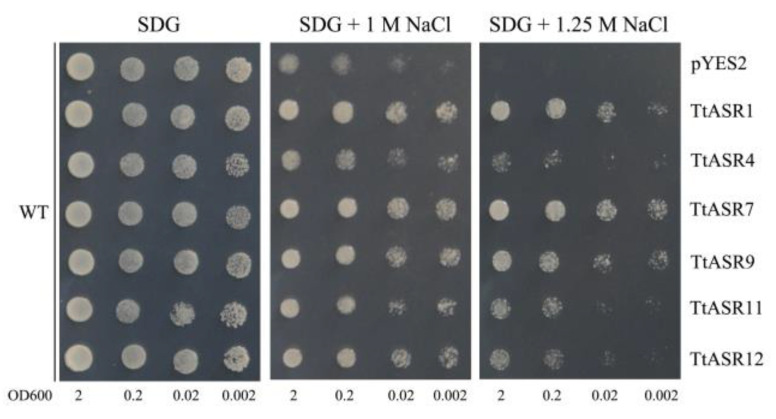
The salt tolerance confirmations of the six candidate *TtASR*s’ expression in yeast. The WT strain transformed with empty vector pYES2 and six different *TtASR-*pYES2 recombinant vectors were spotted on SDG-Ura medium plates supplemented with different concentrations of NaCl (1 M and 1.25 M), 2 μL of each serial dilution (10-fold, from left to right in each panel) of yeast cultures was spotted in turn, and then the plates were incubated for 2 to 5 days at 30 °C.

**Figure 9 ijms-24-15815-f009:**
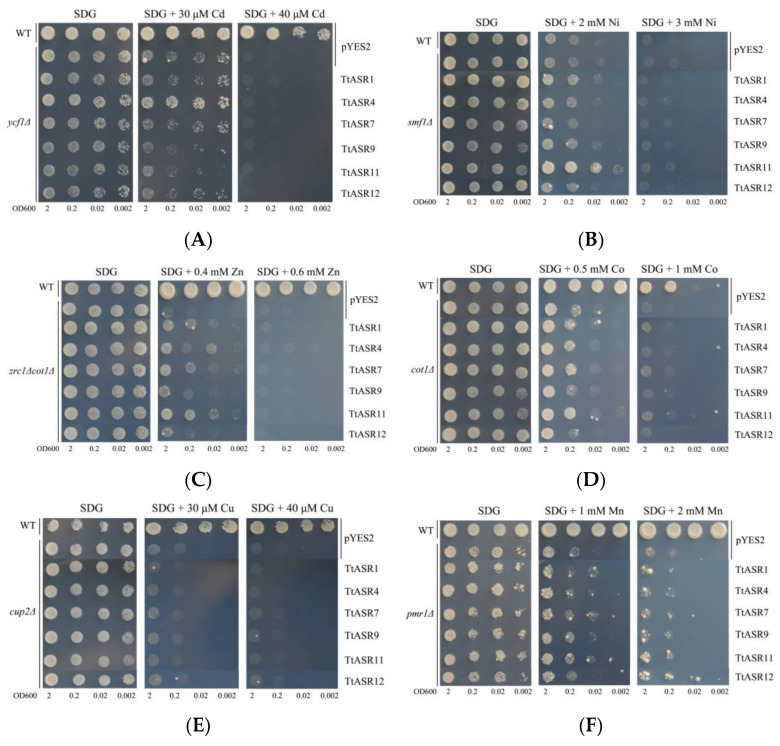
The heavy metal tolerance confirmations of the six *TtASR*s’ expression in yeast. Yeast cultures were adjusted to OD_600_ = 2, and 2 μL of each serial dilution (10-fold, from left to right in each panel) was spotted on SDG-Ura medium plates supplemented with different concentrations of metal stressors. The plates were incubated for 2 to 5 days at 30 °C. (**A**) Cadmium stress tolerance confirmations in yeast mutant strain *ycf1∆*; (**B**) nickel stress tolerance confirmations in yeast mutant strain *smf1∆*; (**C**) zinc stress tolerance confirmations in yeast double-mutant strain *zrc1∆cot1∆*; (**D**) cobalt stress tolerance confirmations in yeast mutant strain *cot1∆*; (**E**) copper stress tolerance confirmations in yeast mutant strain *cup2∆*; (**F**) manganese stress tolerance confirmations in yeast mutant strain *pmr1∆*. The WT strain transformed with pYES2 was used as a positive control, and mutant strains transformed with pYES2 were used as negative controls.

**Table 1 ijms-24-15815-t001:** Nomenclature and subcellular localization of TtASR protein identified from *T. tetragonoides*.

Name	Locus	Protein Length	Major Amino Acids (aa, %)	Mw (kDa)	PI	II	AI	GRAVY	Disordered aa (%)	WoLF_PSORT *	Plant-PLoc
TtASR1	2G0008660.1	106	K(18.9%), A(16.0%), E(15.1%)	12.17	9.20	35.11	47.26	−1.438	35.85	nucl: 14	Nucleus
TtASR2	2G0008680.1	106	K(18.9%), A(15.1%), E(13.2%)	12.10	9.47	33.83	52.74	−1.374	39.62	nucl: 14	Nucleus
TtASR3	2G0008710.1	105	K(18.1%), A(15.2%), E(13.3%)	12.09	9.03	33.69	43.05	−1.473	46.67	nucl: 14	Nucleus
TtASR4	2G0011940.1	107	E(17.8%), K(16.8%), A(14.0%)	11.75	5.81	45.68	50.37	−1.139	56.07	nucl: 7, mito: 5, cyto: 2	Nucleus
TtASR5	2G0011950.1	240	G(21.2%), E(15.0%), T(10.4%)	25.29	5.26	30.51	20.88	−1.369	78.33	nucl: 6, cyto: 4, mito: 1, plas: 1, extr: 1	Cytoplasm
TtASR6	2G0011960.1	119	H(36.1%), A(13.4%), E(11.8%)	13.70	7.03	16.86	43.70	−1.563	55.46	nucl: 5, mito: 4, chlo: 2, cyto: 2	Nucleus
TtASR7	14G0008480.1	105	H(20.0%), A(16.2%), K(16.2%)	12.10	6.59	31.4	47.71	−1.410	44.76	mito: 7, nucl: 3, chlo: 2, cyto: 2	Nucleus
TtASR8	14G0008490.1	107	K(18.7%), E(15.9%), A(13.1%)	12.37	8.98	40.73	52.15	−1.383	37.38	nucl: 14	Nucleus
TtASR9	14G0008500.1	124	H(17.7%), E(17.7%), K(13.7%)	14.30	6.17	47.17	62.90	−1.218	45.16	cyto: 8, nucl: 2, extr: 2, chlo: 1	Nucleus
TtASR10	14G0008560.1	106	K(18.9%), A(16.0%), E(14.2%)	12.13	9.2	34.72	46.32	−1.441	50.94	nucl: 14	Nucleus
TtASR11	14G0011810.1	144	E(20.1%), H(16.7%), A(13.2%)	16.44	6.03	43.16	45.62	−1.287	52.78	nucl: 9, mito: 4	Nucleus
TtASR12	14G0011820.1	277	G(21.7%), E(14.8%), T(10.8%)	29.10	5.12	32.56	16.68	−1.417	81.95	nucl: 11, plas: 1, extr: 1	Cytoplasm
TtASR13	14G0011830.1	107	E(17.8%), K(15.9%), A(12.1%)	11.76	5.81	40.02	54.86	−1.127	53.27	nucl: 7, mito: 5, cyto: 2	Nucleus

* The scores predicted using WoLF PSORT were all listed. MW: molecular weight; PI: isoelectric point; II: instability index; AI: aliphatic index; GRAVY: grand average of hydropathicity. TMHs: transmembrane helices. The molecular weight and isoelectric points of predicted TtASR were detected using the ExPASy proteomics server (https://web.expasy.org/protparam/ (accessed on 29 March 2023)). The contents of disordered amino acids (aa, %) in TtASRs were calculated according to the online program Disordered by Loops/coils definition from DisEMBL 1.5 (Intrinsic Protein Disorder Prediction, http://dis.embl.de/ (accessed on 29 March 2023)). For the subcellular localization prediction, the online programs WoLF_PSORT (https://www.genscript.com/wolf-psort.html (accessed on 29 March 2023)) and Plant-PLoc (http://www.csbio.sjtu.edu.cn/bioinf/plant/ (accessed on 29 March 2023)) were used. WoLF_PSORT predicts the subcellular localization sites of proteins based on both known sorting signal motifs and some correlative sequence features such as amino acid content, and Plant-Ploc mainly predicts plant proteins with 11 subcellular locations in a cell, including: cell wall, chloroplast, cytoplasm, endoplasmic reticulum, extracellular, mitochondrion, nucleus, peroxisome, plasma membrane, plastid, and vacuole.

## Data Availability

Not applicable.

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
