# Peer review of "Identification of the Abscisic Acid-, Stress-, and Ripening-Induced (ASR) Family Involved in the Adaptation of Tetragonia tetragonoides (Pall.) Kuntze to Saline–Alkaline and Drought Habitats"

_ijms, 2023, doi:10.3390/ijms242115815_

Round 1

Reviewer 1 Report

Comments and Suggestions for Authors

In this manuscript, Liu and co-authors present an interesting study that highlights the involvement of ASR family of proteins involved in the adaptation of Tetragonia tetragonoides to saline and alkaline conditions, as well as to drought habitats. The study is timely, being in line with studies performed on species adapted to harsh environments; in this case, the halophyte T. tetragonoides, known to adapt to seawater and drought conditions. The manuscript is coherent and logical, presenting sound and solid data in a comprehensive way.

In this reviewer’s eyes, there is one issue that the authors may want to address/discuss. Specifically, the authors tested the effect of TtASRs expression on salt tolerance in yeast (Figure 8). It would have been interesting to test the effect TtASRs expression in salt-sensitive strains (such as ena1 knock-out), as they did for heavy metal tolerance (Figure 9).

Author Response

Response:

   Thank you for your comments and suggestions concerning our study and manuscript. Actually, in our previous research concerning the identification of halo-tolerant-related genes from tropical littoral halophytes (IJMS. 2018. 19: 3446. doi: 10.3390/ijms19113446), we did choose a yeast salt-sensitive strain, AXT3, which is a three-gene-deletion mutant, including Na+ transporter genes ena1-4, nha1, and nhx1 (doi: 10.1016/s0014-5793(00)01412-5; doi: 10.1371/journal.pone.0137447). We isolated a series of salt-tolerant genes form Ipomoea pes-caprae, but to our surprise, we did not obtain any salt ion transporter genes, just like yeast ena1-4, nha1, and nhx1 (three mutant genes in AXT3). Maybe it came down to the salt-sensitive phenotype changes of AXT3 being less obvious, or the expression levels of transporter genes in plant being relatively low. And in our later research, we used AXT3 only for functional identification of Canavalia rosea sodium/hydrogen (Na+/H+) exchanger (NHX) genes. In this study, the T. tetragonoides' ASR genes encode molecular chaperones or transcription factors, not ion transporters. And possible functions of TtASRs might be protective antioxidant activities, or hydrophilic interaction for maintaining the regular cellular activities under stress challenges. Based on this assumption, we think the wild type yeast be suitable for salt and high osmotic stress tolerance test assay.

Reviewer 2 Report

Comments and Suggestions for Authors

The study focuses on Tetragonia tetragonoides, a halophytic vegetable with food and medicinal uses, prevalent in tropical coastal regions. It explores the ASR gene family within T. tetragonoides, revealing 13 paralogous genes categorized into two subfamilies. The genes are located on two chromosomes, primarily due to segmental duplication. Expression levels of TtASRs indicate their involvement in responding to abiotic stressors. Functional identification through a yeast expression system suggests TtASRs' significant roles in adapting to saline-alkaline soils and drought stress. This research enhances our comprehension of ASRs in halophyte adaptation and plant ASR protein evolution.

The manuscript demonstrates meticulous research on T. tetragonoides' ASR gene family, offering valuable insights into its role in halophyte adaptation to extreme environments and plant ASR protein evolution.

Author Response

Response:

Thanks so much for your comments. We do hope that our manuscript would meet your requirements.
